Knowledge of malaria prevention among pregnant women and female caregivers of under-five children in rural southwest Nigeria

Adebayo Ayodeji M. 1
Akinyemi Oluwaseun O. 2 seunakinyemi@hotmail.com
Cadmus Eniola O. 1
1 Department of Preventive Medicine and Primary Care, College of Medicine, University of Ibadan , Ibadan , Nigeria
2 Department of Health Policy and Management, College of Medicine, University of Ibadan , Ibadan , Nigeria
Erez Offer
Electronic publication date: 2015 Feb 24
Publication date: 2015
Volume: 3
Electronic Location ID: e792
Received 2014 Nov 19; Accepted 2015 Feb 4
Copyright: © 2015 Adebayo et al.
Copyright year: 2015
Copyright holder: Adebayo et al.
License: This is an open access article distributed under the terms of the Creative Commons Attribution License, which permits unrestricted use, distribution, reproduction and adaptation in any medium and for any purpose provided that it is properly attributed. For attribution, the original author(s), title, publication source (PeerJ) and either DOI or URL of the article must be cited.
License URL: https://creativecommons.org/licenses/by/4.0/

Keywords: Under-five caregivers, Malaria prevention, Pregnant women, Nigeria, Insecticide treated net, Rural dwellers

Funding: National Primary Health Care Development Agency (NPHCDA) This study was funded by the National Primary Health Care Development Agency (NPHCDA), Abuja, Nigeria. The funders had no role in study design, data collection and analysis, decision to publish, or preparation of the manuscript.

==============================
Introduction. The morbidity and mortality from malaria are still unacceptably high in the developing countries, especially among the vulnerable groups like pregnant women and under-five children, despite all control efforts. The knowledge about the preventive measures of malaria is an important preceding factor for the acceptance and use of malaria preventive measures like Insecticide Treated Nets (ITN) by community members. Therefore, this study assessed the knowledge of malaria prevention among caregivers of under-five children and pregnant women in a rural community in Southwest Nigeria.

Methodology. This is part of a larger malaria prevention study in rural Southwest Nigeria. A descriptive cross-sectional survey was conducted among pregnant women and caregivers of under-five children in Igbo-Ora, a rural town in Southwest Nigeria using a semi-structured, interviewer-administered questionnaire. Information was obtained on knowledge of malaria prevention, and overall composite scores were computed for knowledge of malaria prevention and ITN use. Data were analyzed using SPSS version 16. Associations between variables were tested using a Chi-square with the level of statistical significance set at 5%.

Results. Of the 631 respondents, 84.9% were caregivers of under-five children and 67.7% were married. Mean age was 27.7 ± 6.3 years with 53.4% aged between 20 and 29 years. Majority (91.1%) had at least primary school education and 60.2% were traders. Overall, 57.7% had poor knowledge of malaria prevention. A good proportion (83.5%) were aware of the use of ITN for malaria prevention while 30.6% had poor knowledge of its use. Respondents who were younger (<30 years), had at least primary education and earn <10,000/per month had significantly poor knowledge of ITN use in malaria prevention. Majority (60.0%) respondents had poor attitude regarding use of ITNs.

Conclusion. This study showed that the knowledge of malaria prevention is still low among under-five caregivers and pregnant women in rural Southwest Nigeria despite current control measures. There is a need for concerted health education intervention to improve the knowledge of rural dwellers regarding malaria prevention, including the use of ITN. This will go a long way to improving the reported low level of ownership and utilization of ITN in the rural areas.

Introduction

Malaria is a completely preventable disease; however, about 3.4 billion people are at risk of the disease globally with 1.2 billion people at high risk (World Health Organization, 2013). In 2012, malaria was responsible for the death of approximately 482,000 under-five children even though an estimated 136 million Insecticide Treated Nets (ITNs) were distributed to endemic countries the same year (World Health Organization, 2013). Thus, malaria is still a major public health concern particularly in sub-Saharan Africa and other parts of the developing world (Pluess et al., 2009; Pluess et al., 2010). In Nigeria, malaria is responsible for around 60% of the out-patient visits to health facilities, 30% of childhood death, 25% of death in children under one year and 11% of maternal deaths (National Population Commission, 2008; Noland et al., 2014). Similarly, about 70% of pregnant women suffer from malaria, which contributes to maternal anemia, low birth weight, still births, abortions and other pregnancy-related complications (Federal Ministry of Health Abuja, 2005). The financial loss due to malaria is estimated to be about 132 billion Naira ($797 million) annually in form of treatment costs, prevention costs and loss of man-hours (Noland et al., 2014; World Health Organization, 2012).

Malaria, a debilitating febrile and life threatening illness, is caused by a parasite called Plasmodium. Its route of transmission still remains as bites from infected female anopheles mosquitoes. Environmental factors and behavioral patterns of vectors and human populations combine to provide favorable conditions for malaria transmission (Boutin et al., 2005). Proven effective options to reduce morbidity and mortality include early diagnosis, combined with prompt effective therapy and malaria prevention through reduction of human-vector contact, especially with the use of ITNs (World Health Organization, 2007). Perceptions about malaria illness, particularly households’ perceived susceptibility and beliefs about the seriousness of the disease, are important preceding factors for decision-making concerning preventive and curative actions (Rakhshani et al., 2003). The understanding of the possible causes, modes of transmission, and individual preference and decision-making about the adoption of preventive and control measures vary from community to community and among individual households (Adongo, Kirkwood & Kendall, 2005). There have been a considerable number of reports about knowledge, attitudes, and practices relating to malaria and its control from different parts of Africa. These reports concluded that misconceptions concerning malaria still exist and that practices for the control of malaria have been unsatisfactory (Ahorlu et al., 1997; Alaii et al., 2003; Laver, Wetzels & Behrens, 2001; Obol, David Lagoro & Christopher Garimoi, 2011).

Achieving sustainable control of the disease depend on extensive public health promotional programs which focus on current and proven methods of malaria prevention and management. While much is known about vector biology and behaviour and the malaria parasites, the importance of human behaviour in malaria transmission has not been critically evaluated. Studies focusing on the current practices of malaria prevention and treatment options in the population are sparse. Thus, it is expedient to evaluate current knowledge of malaria prevention practices and management options as well as the uptake of the management options. In most high-burden countries (including Nigeria), ITN coverage is still below agreed targets (Minja & Obrist, 2005). This may be related to the perception of its use among the community members. The knowledge about the preventive measures of malaria is an important preceding factor for the acceptance and use of ITN for malaria control by the community members (Minja et al., 2001). Therefore, this study assessed the knowledge of malaria prevention with emphasis on knowledge of ITN use among pregnant women and caregivers of under-five children in Igbo-Ora.

Igbo-Ora is a rural community in Ibarapa Central Local Government Area (LGA) of Oyo State, Southwest Nigeria. The main towns in the LGA are Igbo-Ora and Idere. Igbo-Ora is the larger town with a population of 60,000 people. The LGA has ten political wards with seven located in Igbo-Ora, the study site. Igbo-Ora is located about 128 km from Lagos—the Nigerian economic capital. The predominant language in the study area is Yoruba. However, migrant farm labourers from the republic of Benin, Togo and Ghana reside in the LGA. There are some nomadic Fulanis who live in settlements around the town.

Igbo-Ora is further divided into six census areas comprised of 62 enumeration areas, each with an average population of 600 people. Furthermore, each enumeration area is divided into compounds, and each compound has about 100 women in the reproductive age group.

Methodology

The study population is made up of pregnant women in their reproductive age (15–49 years) and female caregivers of under-five children, who have lived in the community for at least one year. For the purpose of this study, caregivers may include mothers and female guardians of under-five children.

A community-based descriptive cross-sectional survey was carried out using a multistage cluster sampling technique. Three enumeration areas were selected by simple random sampling through balloting from each of the six census areas. Subsequently, two compounds were selected from each of these enumeration areas by balloting. A minimum sample size of 126 was estimated using the Leslie and Kish formula for approximating sample size for cross sectional study. This was multiplied by a factor of two to adjust for clustering effect. A total of 631 eligible and consenting caregivers of under-five children and pregnant women in the households within the selected compounds were then interviewed. All the people who were approached consented to participate in the study.

A pre-tested semi-structured interviewer-administered questionnaire was used to collect information on the socio-demographic characteristics of respondents, knowledge of malaria prevention and knowledge of ITN use, and perception of ITN use in malaria prevention. Knowledge about malaria prevention and ITN use were assessed through a six-point score each. A point was given to each correct answer, and 0 to each wrong answer. The mean knowledge score was used to dichotomize knowledge scores. Knowledge about malaria prevention was categorized into “good” (4–6) and “poor” (0–3) while scores of 0–4 were deemed poor and 5–6 were considered good for knowledge of ITN use for malaria prevention. The questionnaire was translated to Yoruba in order to enable proper understanding by respondents and back-translated to English to ensure that the original meaning was retained. Questionnaire administration was done by four trained research assistants whose minimum educational qualifications were to have a National Certificate in Education (NCE) or Ordinary National Diploma (OND), and were fluent in speaking the local dialect. The questionnaire was pre-tested in Idere, the second main town in Ibarapa Central LGA.

Data were analyzed using the Statistical Package for Social Sciences (SPSS version 16). Associations between variables were tested using Chi-square with the level of statistical significance set at 5%. The raw data for this study may be accessed through: https://docs.google.com/a/cartafrica.org/file/d/0B62N6G90gxJZNzdIT05NQ2w1QWs/edit.

The Oyo State Ethical Review Committee, Ministry of Health, Ibadan gave ethical clearance and approval for this study (reference number: AD 13/479/76). Informed verbal consent was also gotten from individual research participant before data collection. The respondents were also reminded of their right to decline to take part in the study as well as to withdraw any time during the interview. Confidentiality was assured and maintained throughout the study. The importance of the study was explained to participants as well as how their participation in the study will contribute towards malaria prevention programing in Nigeria.

Results

Figure 1 shows the distribution of respondents. A total of 631 respondents were interviewed, out of whom 536 (84.9%) were caregivers of under-five children. The socio-demographic characteristics of respondents are shown in Table 1. The mean age of respondents was 27.7 ± 6.3 years, with about 53% aged between 20 and 29 years. More than 90% have at least primary education while trading (60.2%) was the most common occupation among respondents. About 70% were married and living with their spouses. A greater proportion, (71.9%) of respondents were either in or from a monogamous relationship. About 60% of respondents have 2 children or less while almost all (97.5%) have 2 or less under-five children. About 97% were from the Yoruba ethnic affiliation and more than half, (57.6%) were Muslims. Most respondents, 64.7% earned less than ₦10,000 ($59.4) monthly. Only 8.4% earned ₦30,000 ($178.2) and above. The median average income was ₦5000 ($29.7); range: ₦1000–₦150,000.

Figure 1 Distribution of respondents.

Table 1 Socio-demographic characteristics of the respondents (N = 631).

Socio-demographic characteristics	Frequency	Percentage	
Age group			
<20	36	5.7	
20–29	337	53.4	
30–39	230	36.5	
≥40	28	4.4	
Level of education			
No formal education	56	8.9	
Primary	206	32.6	
Secondary	275	43.6	
Tertiary	94	14.9	
Occupation			
Trading	380	60.2	
Civil servant	78	12.4	
Farming	15	2.4	
Unemployed	158	25	
Income (₦)			
<1,000	408	64.7	
10,000–19,000	107	17.0	
20,000 – 29,000	63	10.0	
≥30,000	53	8.4	
Marital status			
Single	30	4.8	
Co-habiting	85	13.5	
Married, living together	440	69.7	
Married, living alone	67	10.6	
Separated	8	1.3	
Divorced	1	0.2	
Family type			
Monogamous	454	71.9	
Polygamous	177	28.1	
Total number of children			
0–2	380	60.2	
3–4	228	36.1	
≥5	23	3.7	
Number of U-5 children			
0–2	615	97.5	
3–4	16	2.5	
Ethnic group			
Yoruba	611	96.8	
Igbo	7	1.1	
Hausa	5	0.8	
Others	8	1.3	
Religion			
Christianity	268	42.5	
Islam	363	57.6	

Table 2 revealed that less than half of respondents (42.3%) had good knowledge of malaria prevention. Table 3 shows that respondents’ knowledge regarding methods of malaria prevention. About 85%, 82%, and 75% knew that malaria could be prevented through keeping the environment clean, clearing of bushes around houses and use of ITN respectively. However, approximately 77% and 82% of respondents felt taking native concoction and using malaria prophylaxis respectively could prevent the occurrence of malaria infection.

Table 2 Respondents’ knowledge score of malaria prevention.

Knowledge score (0–6)	Frequency	Percentage	
Good (4–6)	269	42.3	
Poor (0–3)	364	57.7	
Notes.

Mean=3.4 ± 1.2.

Table 3 Respondents’ distribution of knowledge about methods of malaria prevention.*

Methods of malaria prevention	Yes (%)	No (%)	
Drinking native concoction	485 (76.9)	146 (23.1)	
Keeping our environment clean	534 (84.6)	97 (15.4)	
Clearing of bushes around the house	516 (81.8)	115 (18.2)	
Use of antibiotics	300 (47.5)	331 (52.5)	
Use of ITN	473 (75.0)	158 (25.0)	
Use of antimalarial prophylaxis	519 (82.3)	112 (17.7)	
Notes.

* Multiple response.

Nearly 84% of participants had heard of ITN, while almost 70% had good knowledge of the use of ITN in malaria prevention (Table 4). Regarding the use of ITN in malaria prevention, about 80% knew that ITN is useful in malaria prevention and over three-quarters understood that it could kill mosquito (Table 5).

Table 4 Respondents’ awareness of ITN and knowledge score of ITN use in malaria prevention.

	Number	Percent	
Awareness			
Yes	527	83.5	
No	104	16.5	
Knowledge score of ITN use in malaria (0–6)			
Good (5–6)	438	69.4	
Poor (0–4)	193	30.6	
Total	631	100.0	
Notes.

Mean knowledge score = 4.4 ± 1.7.

Table 5 Respondents’ distribution of knowledge about the use of ITN in malaria prevention.*

Knowledge if ITN use	Yes (%)	No (%)	
Keep flies away	409 (64.8)	222 (35.2)	
Keep rats away	171 (27.1)	460 (72.9)	
Fishing	152 (24.1)	479 (75.9)	
Prevents mosquito bite	520 (82.4)	111 (17.6)	
Useful in malaria prevention	503 (79.7)	128 (20.3)	
Kills mosquitoes	476 (75.4)	155 (24.6)	
Notes.

* Multiple response.

Table 6 shows the attitude of under-five caregivers and pregnant women regarding ITN and its use. Roughly 63% of respondents did not agree that window/door nets was better or the same as ITN. Similarly, 61% of participants disagreed with the statement that “ITN does not make any difference in malaria prevention.” Well over half of participants (56.1%) agreed that “either one uses mosquito net or not, those that will be infected with malaria will definitely have it.” Majority were indifferent as to whether ITN causes irritation, heat rash, cough, vomiting in pregnancy, miscarriage, mortality in children and bad odor.

Table 6 Respondents’ attitude towards ITN and its use.

	Agree (%)	Don’t know (%)	Disagree (%)	
It is the same as window/door net	97 (15.4)	134 (21.2)	400 (63.4)	
It does not make any difference in malaria prevention	103 (16.3)	144 (22.8)	384 (60.9)	
ITN smells badly	43 (6.8)	451 (71.5)	137 (21.7)	
ITN causes irritation	20 (3.2)	469 (74.3)	142 (22.5)	
ITN causes heat rashes	21 (3.3)	466 (73.9)	144 (22.8)	
ITN causes cough/illness	10 (1.6)	465 (73.7)	156 (24.7)	
ITN causes nightmares/bad dreams	5 (0.8)	459 (72.7)	167 (26.5)	
The chemical in ITN can kill children	3 (0.5)	450 (71.4)	178 (28.2)	
ITN use in pregnancy can cause miscarriage	2 (0.3)	457 (72.4)	172 (27.3)	
ITN can cause vomiting in pregnant women	10 (1.6)	464 (73.5)	157 (24.9)	
ITN cannot kill mosquitoes	26 (4.1)	338 (53.6)	267 (42.3)	
ITN is not readily available	137 (21.7)	389 (61.6)	105 (16.6)	
ITN is expensive	157 (24.9)	419 (66.4)	55 (8.7)	
Window nets/door nets are better	35 (5.5)	212 (33.6)	384 (60.9)	
Ordinary net (without insecticide) is better/preferable	40 (6.3)	202 (32.0)	389 (61.6)	
Either you use mosquito net or not those who will have malaria will still have it	354 (56.1)	121 (19.2)	156 (24.7)	

Table 7 demonstrated that mothers and under-five caregivers with higher educational status, who were civil servants, with higher income, and married had good knowledge of malaria prevention (p<0.05). Similarly respondents who earned high income and were married had significantly better knowledge of ITN use (Table 8).

Table 7 Association between respondent characteristics and knowledge of malaria prevention.

	Knowledge of Malaria prevention			
Variables	Poor n(%)	Good n(%)	Chi-square	p-value	
Level of education					
No formal education	33 (58.9)	23 (41.1)			
Primary	131 (63.6)	75 (36.4)	23.145	<0.001	
Secondary	165 (60.0)	110 (40.0)			
Tertiary	33 (35.1)	61 (64.9)			
Occupation					
Trading	232 (61.1)	148 (38.9)			
Artisan	34 (50.7)	33 (49.3)	35.317	<0.001	
Civil servant	24 (30.8)	54 (69.2)			
Farming	14 (93.3)	1 (6.7)			
Others	58 (63.7)	33 (36.3)			
Average monthly income					
<10,000	249 (61.0)	159 (39.0)			
10,000–19,999	60 (56.1)	47 (43.9)	10.813	0.013	
20,000–29,999	25 (39.7)	38 (60.3)			
≥30,000	28 (52.8)	25 (47.2)			
Marital status					
Never married	86 (74.8)	29 (25.2)	17.436	<0.001	
Ever married	276 (53.5)	240 (46.5)			
Religion					
Christianity	142 (53.0)	126 (47.0)	3.661	0.056	
Islam	220 (60.6)	143 (39.4)			
Respondents					
Pregnant	42 (64.6)	23 (35.4)			
Under-5-caregiver	310 (57.8)	226 (42.2)	5.128	0.077	
Both	12 (40.0)	18 (60.0)			

Table 8 Association between respondent characteristics and knowledge of ITN use.

	Knowledge of utilization of ITN			
Characteristics	Poor n(%)	Good n(%)	X 2	p value	
Level of education					
No formal education	20 (35.7)	36 (64.3)			
Primary	74 (35.9)	132 (64.1)	19.504	<0.001	
Secondary	88 (32.0)	187 (68.0)			
Tertiary	11 (11.7)	83 (88.3)			
Occupation					
Trading	128 (33.7)	252 (66.3)			
Artisan	20 (29.9)	47 (70.1)			
Civil servant	10 (12.8)	68 (87.2)	13.627	0.009	
Farming	5 (33.3)	10 (66.7)			
Others	30 (33.0)	61 (67.0)			
Income					
<10,000	149 (36.5)	259 (63.5)			
10,000–19,999	27 (25.2)	80 (74.8)	13.452	0.004	
20,000–29,999	9 (14.3)	54 (85.7)			
≥30,000	8 (15.1)	45 (84.9)			
Marital status					
Never married	66 (57.4)	49 (42.6)	47.592	<0.001	
Ever married	127 (24.6)	389 (75.4)			
Religion					
Christianity	76 (28.4)	192 (71.6)	1.089	0.169	
Islam	117 (32.2)	246 (67.8)			
Respondents					
Pregnant	14 (21.5)	51 (78.5)			
Under-5-caregiver	173 (32.3)	363 (67.7)	4.811	0.090	
Both	6 (20.0)	24 (80.0)			

Discussion

This study was conducted to assess knowledge of malaria prevention with emphasis on ITN use among female caregivers of under-five children and pregnant women in Igbo-Ora, a rural community in Southwest Nigeria. The overall knowledge of malaria prevention practices among majority of the respondents was found to be poor. This finding agrees with the submissions of Fawole & Onyeaso (2008) who showed that even among health workers in Ibadan, Southwest Nigeria, knowledge of malaria preventive strategies was poor. However, this finding is at variance with the conclusions of Adegun, Adegboyega & Awosusi (2011) and Oyewole & Ibidapo (2007) who showed that the general knowledge about malaria prevention among urban residents in Southwest Nigeria was good. The knowledge picture seen among respondents in this study could be as a result of a lack of exposure to health education messages regarding malaria prevention, being rural dwellers, or the respondents’ poor health seeking behavior. For those who had exposure to health messages in health facilities, some might find it difficult to understand malaria prevention information given during antenatal and postnatal clinics. This might be partly due to inappropriate means of communication and delivery of these messages by the health workers in addition to the low level of education of the respondents. The respondents’ level of education was found to be significantly associated with knowledge of malaria prevention. Therefore, achieving malaria prevention, just like any other health message, depend on the level of education of respondents among other reasons.

Furthermore, relatively high proportions of the respondents in our study knew that use of antimalarial prophylaxis in pregnancy, clearing of bushes around the house and keeping the environment clean were part of malaria prevention strategies. The knowledge of the importance of prophylaxis for malaria prevention may be a function of the fact that they had benefited from intermittent preventive treatment (IPT) during index or previous pregnancies during which they might have had health education on the issue. However, other studies from Nigeria (Falade et al., 2006; Okeke, Uzochukwu & Okafor, 2006) and Tanzania (Comoro et al., 2003) have documented that gaps still exist in the knowledge of causation and treatment of malaria in rural areas and that these gaps have serious public health implications.

Most of these rural respondents still exhibited some myths and misconceptions about malaria prevention. Drinking of native concoction and use of antibiotics for malaria treatment were some of the erroneous believes reported. The finding on the use native concoction by pregnant women in Nigeria is corroborated by Fakeye and colleagues (2009) who advised that questions on herbal drug use should be routinely asked by health workers during antenatal care to forestall dangerous drug interactions. Herbal preparations, though not medically recommended for the treatment of diseases like malaria is an innate traditional practice which is considered normal by people because of a deep cultural attachment. Inadequate information from health workers and the respondents’ low level of education could also contribute to the misconceptions of malaria treatment. Continuous efforts at providing necessary information by relevant health organizations are needed to control and prevent incidence of malaria in the general public.

More than two-thirds of the respondents had good knowledge of ITN use for malaria prevention. Awosan et al. (2013), in a study to determine the prevalence and barriers to the use of insecticide-treated nets among pregnant women attending ante-natal clinic in a Specialist Hospital at Sokoto, reported similar finding. However, in a hospital-based study about awareness of ITN use in Abeokuta, Southwest Nigeria, Runsewe-Abiodun, Iyabo & Christy (2012) reported low awareness and linked this to the low utilization among the pregnant women interviewed. The current ongoing aggressive public health campaign which involves mass distribution of ITNs at PHC centers could account for the high awareness noted in this study. However, some respondents had poor attitudes and misconceptions about the use of ITNs. The respondents’ attitudes may have a strong implication on ownership and utilization of ITNs (Onwujekwe et al., 2005).

Conclusion and recommendations

Despite concerted efforts at malaria control nationwide, knowledge of malaria prevention was below average among rural caregivers of under-five children and pregnant women in the study area. Myths and misconceptions about malaria prevention is still prevalent. There is a need for a concerted health education intervention to improve the knowledge of rural dwellers regarding malaria prevention especially with the use of Insecticide Treated Net. Continuous efforts at providing necessary information by relevant health organizations are needed to control and reduce incidence of malaria in the general public.

Additional Information and Declarations

Competing Interests

Author Contributions

Human Ethics

Data Deposition

The authors declare there are no competing interests.

Ayodeji M. Adebayo and Oluwaseun O. Akinyemi conceived and designed the study, performed the study, analyzed the data, contributed analysis tools, wrote the paper, prepared figures and/or tables, reviewed drafts of the paper.

Eniola O. Cadmus performed the study, contributed analysis tools, reviewed drafts of the paper.

The following information was supplied relating to ethical approvals (i.e., approving body and any reference numbers):

Oyo State Ethical Review Committee, Ministry of Health, Ibadan, Nigeria. Reference number: AD 13/479/76

The following information was supplied regarding the deposition of related data:

The dataset for this study is accessible via Google drive: https://docs.google.com/a/cartafrica.org/file/d/0B62N6G90gxJZNzdIT05NQ2w1QWs/edit.

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
