# Peer review of "Knowledge of malaria prevention among pregnant women and female caregivers of under-five children in rural southwest Nigeria"

_PeerJ, doi:10.7717/peerj.792_

## Round 0.1 · original submission · Major Revisions

The authors are presenting an important subject, yet the reviewers both raised some major reservations that need to be addressed before this manuscript can be reconsidered for publication

Reviewer 1 ·

Basic reporting

The article is descriptive survey, addressing an important issue concerning knowledge of Malaria prevention among pregnant women and female caregivers of under-5 children in rural parts of Nigeria .
The Introduction section focuses mainly on relevant data concerning Malaria and it's health social and economic burden on society, as well as the benefits of prevention and the importance of awareness to prevention means through public health education. It is well written.
I would consider moving the background information about the area and the division into smaller units (through census etc. to compounds) to the introduction section instead of the Methodology section, as I understand this is a general division, not one that is done especially for the purpose of the study (lines 74-84 in PDF for review). The information about the random selection of compounds for the present study is indeed relevant and should be left in Methodology section.

Experimental design

The Methodology section lacks specific information regarding the questionnaire and scoring system used for the study, defining which scores are given the definition of "good" versus "poor" knowledge etc. The lack of this information is apparent later in the result section when in certain tables (for example table 2) the table presents percentage of good vs. poor knowledge, while the caption below the table states the mean score in numbers – since the reader did not receive information about the scoring system (what is the maximum score? Which score divides knowledge into good vs poor? Etc.) this mean number has no relevance to the reader.
See also comments regarding the "Results" and "discussion" sections, adressing the same issue.

Validity of the findings

The Result section –
• The representation of the" income groups" data in Figure 2 does not add significant information as opposed to adding this data to Table 1 and leaving the text addressing the differences in income.
• Table 2 & table 4 – as explained above – the mean score in the caption does not help the reader relate to which score is goodqpoor – I would again advise relating to it in methodology section and also in the table (for example writing "poor 0-3" etc.)
• In Table 7:Association between respondent characteristics 316 and knowledge of malaria prevention and Table 8: Association between respondent characteristics and knowledge of ITN use – both tables show only statistically significant results (and therefore even between the 2 tables the categories represented differ) - I would advise using the same categories in both tables, even if not statistically significant, so the reader will have a notion as to which categories were studied for association. Highlighting can be used for statistically significant data, in addition to addressing it in the text.
• Line 148 – should be "better knowledge" ? again when using "good" as the result parameter here when you say "significantly good knowledge" it is not clear if you use it as a score or as a word (in that case the word better is indeed more appropriate).

Discussion section –
• The main finding line 153-154 in the PDF is cited as – "The overall knowledge of malaria prevention practices among the respondents was found to be below average" – when saying such a statement one must understand what the authors consider "average" knowledge in this area. Table 2 shows that over 40% of the responders had "good" overall knowledge of malaria prevention – is this considered below average?
• Same comment as for line 189 –" Knowledge of ITN use for malaria prevention was above average" – what is the expected average knowledge? Do the authors compare this data to other studies addressing this issue? To their preliminary pre-tested questionnaire? To a score they have used in the study?
• Overall the discussion is very interesting, and addresses important issues such as differences according to level of education as an issue for public health strategies, and the importance of awareness to different means of malaria prevention.

Additional comments

Thank you for addressing this important issue.
Additional proofreading of the article might be advisable.

·

Basic reporting

About the basic reporting section, I found it good and well organized.
Few minor comments:
In the ABSTRACT introduction you are presenting the investigated groups "Under 5-caregivers and pregnant women". I think it would be better specified that you are talking about caregivers of children under 5 years old.
It is also not clear who are those caregivers- parents? Or maybe personal, working in a kindergarten.
In the introduction section, rows 61-69 there is much unnecessary replication that can be summarized in one sentence.
But it's lacking information about malaria prevention and treatment that I find relevant to this article.
The tables and graphs are well arranged and clarify the text very nicely.

Experimental design

In the methodology section it wasn't clear enough what the research framework is (health clinic, school or other).
This section is also lacking information about what was the total number of people addressed, from which 631 responded and how were they addressed at.

Validity of the findings

In the discussion section I would expect a discussion about a possible bias in the responders group. Those who agreed to participate may be of more experience and knowledge about malaria disease and prevention.
You also state in rows 165-168 that responders level of education was found to be significantly associated with knowledge of malaria prevention , but table one shows that only minority of responders (8.9%) has no formal education, which is a possible bias for your research.
Yet table 7 fails to prove your point about level of education and knowledge of malaria prevention.

Additional comments

I would like to thank you for the opportunity reviewing this article.
Obviously an important issue is addressed here by the investigators.
Yet there are few minor corrections to be made to clarify the study and its implications.
I also find it important to understand what kind of intervention was done during the interview or afterwards in the population accessed.And I would like to read about more specified suggestions in the "conclusion and recommendation" section for health education intervention.

---

## Round 0.2 · accepted · Accept

I would like to congratulate the authors for addressing the concernrs raised by the reviewers the manuscript can now be accepted for publication